# Association between physical activity level and cardiovascular disease: An empirical analysis based on CHARLS data in 2018

Haiwei Li , Liang Luo*, Jing Zhang, Yanhao Zhao, Peipei Cheng, Dan Liu, Liwei Guo

School of Physical Education, Shanxi Normal University, Taiyuan, China

* 223114030@sxnu.edu.cn

## Abstract

### Objective

Cardiovascular disease (CVD) was a global public health challenge. This study aimed to investigate the association between physical activity level (PAL) and CVD among middle-aged and older adults in China, to provide evidence to inform strategies for CVD prevention and management.

### Methods

The diagnosis of CVD was based on self-reported physician-diagnosed heart disease or stroke, while PAL were classified using calculations derived from a structured questionnaire. Both datasets were obtained from the 2018 China Health and Retirement Longitudinal Study (CHARLS) survey. A multivariate logistic regression model was used for the primary correlation analysis. Additionally, restricted cubic spline (RCS) regression was employed to examine the potential nonlinear association between PAL and CVD.

### Results

The final analysis included 9,015 participants, 1,069 of whom were diagnosed with CVD, yielding a prevalence of 11.86%. After adjusting for all covariates, the multivariate-adjusted odds ratios (ORs) for the moderate PAL group (600−3000 MET-minutes/week) and the high PAL group (>3000 MET-minutes/week) were 0.79 (95% CI: 0.64–0.97) and 0.72 (95% CI: 0.60–0.87), respectively, compared to the low PAL group (<600 MET-minutes/week). Furthermore, restricted cubic spline analysis revealed a significant linear relationship between PAL and CVD (nonlinear P > 0.05), indicating that the OR for CVD decreased with higher levels of PAL.

**Data availability statement:** All relevant data are within the manuscript and its Supporting Information files.

**Funding:** HL was supported by the Shanxi Province Philosophy and Social Science Planning Project (No. 2020YY080). The funder had no role in the study design, data collection and analysis, decision to publish, or preparation of the manuscript.

**Competing interests:** The authors have declared that no competing interests exist.

## Conclusions

This study revealed a negative correlation between PAL and the prevalence of CVD. Furthermore, our findings suggested that middle-aged and older adults should maintain moderate to high levels of physical activity, as this was associated with a lower risk of CVD.

## 1. Introduction

Cardiovascular diseases (CVD) were the leading cause of mortality worldwide [1]. A study predicted that, assuming risk factors remain constant, the annual number of cardiovascular events increase by more than 50% between 2010 and 2030 due solely to population aging and growth [2]. According to the 2019 update from the American Heart Association on heart disease and stroke statistics, the prevalence of CVD was 25.1% among men and 17.6% among women aged 60–80 years. In the 80+ age group, the average prevalence rose to 43.3% for men and 34.3% for women [3]. Aging had become a major global concern, with the burden of CVD primarily affecting older adults [4].

Previous studies have identified several risk factors for CVD, including hypertension, diabetes, obesity, dyslipidemia, smoking, a sedentary lifestyle, and insufficient physical activity [5]. Most CVD was preventable through lifestyle modifications that address key behavioral risk factors, such as smoking, obesity, and physical inactivity [6]. Urbanization, aging, and digitization have significantly contributed to the decline in per capita PAL. Indeed, physical inactivity was currently the fourth leading cause of death globally [7]. underscoring the critical importance of maintaining an active lifestyle. Regular physical activity can reduce oxidative stress and systemic inflammation, two major mechanisms underlying CVD. By lowering levels of inflammatory markers such as CRP, TNF-$\alpha$, INF-$\gamma$, and NF-$\kappa$B, and increasing anti-inflammatory cytokines like IL-10, IL-4, and IL-8, physical activity can help prevent the onset and progression of CVD, particularly among middle-aged adults [8].

A substantial body of epidemiological evidence demonstrates an independent inverse correlation between PAL and both CVD mortality and all-cause mortality [9–11]. Research shows that, compared to participants reporting no leisure-time physical activity, those meeting the minimum physical activity level recommended by the 2008 U.S. guidelines (500 MET-minutes/week) were associated with a lower CVD risk. In contrast, physical activity levels at twice the minimum guideline – recommended level – 1,000 MET – minutes/week and four times the level – 2,000 MET – minutes/week showed an association with substantially lower CVD risk [12]. Moreover, a study by Thompson D *et al.*[13], recommended that middle-aged and older men in the UK engage in 3,000–6,000 MET-minutes/week of physical activity. Similarly, a study by Bakker EA *et al.* [14], observed that when physical activity exceeded 7,258 MET-minutes/week among adults in northern Netherlands, it was associated with a lower risk of adverse outcomes in patients with cardiovascular disease. While these findings provided valuable insights into the relationship between PAL and CVD,

limited information was available regarding this relationship in China, which has the largest proportion of older adults and was undergoing the most rapid aging. Therefore, the objective of this study was to examine the association between PAL and CVD in a nationally representative cohort using data from the 2018 CHARLS.

## 2. Materials and methods

### 2.1 Study design and participants

This cross-sectional study utilized data from the 2018 (wave 4) of the CHARLS [15], which included an initial sample of 19,816 individuals. The original CHARLS survey data were approved by the Ethics Review Committee of Peking University (No. IRB00001052–11015), and all participants provided informed consent. All procedures involving human subjects adhered to the guidelines of the institutional research committee and followed the principles outlined in the 1964 Declaration of Helsinki [16]. Several exclusion criteria were applied, resulting in the removal of participants who met the following conditions: (1) absence of annual household consumption information (n = 8,272); (2) missing data on age, gender, education, residence, marital status, social activities, depression, or physical activity (n = 2,374); (3) age below 45 years (n = 113); and (4) residence in a special area (n = 42). Ultimately, the study included a total of 9,015 participants (see Fig 1).

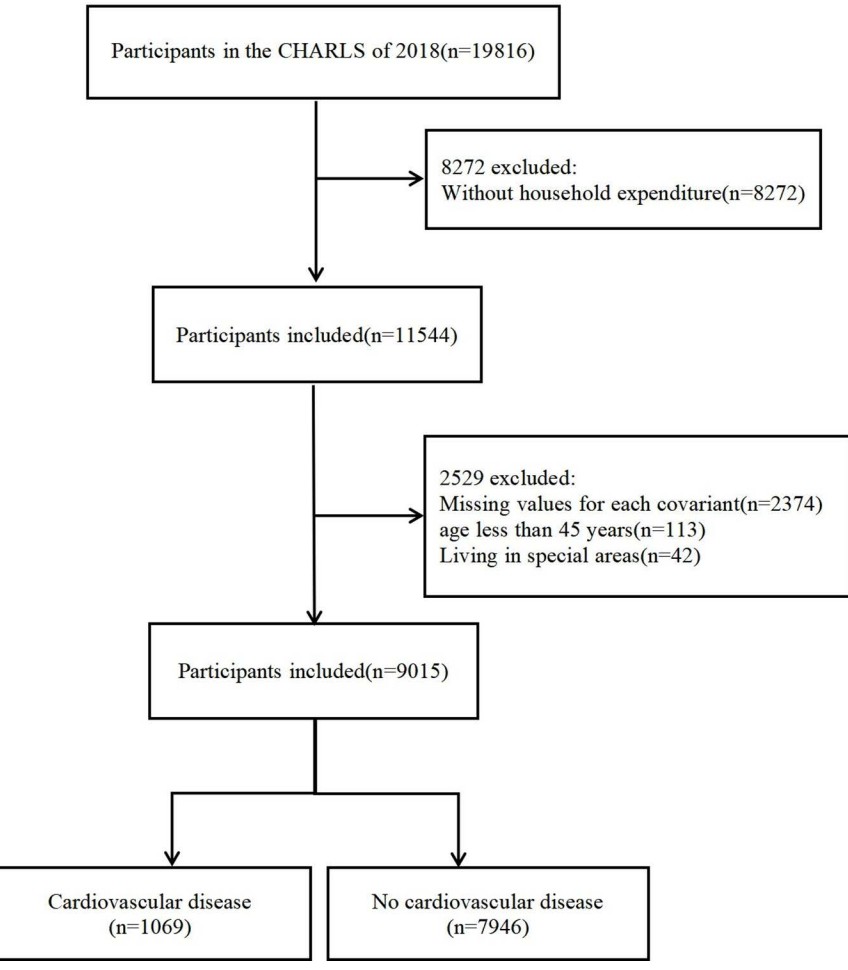

**Fig 1. Flow diagram of the participants screening procedures.**

## 2.2 Exposure variable and outcomes variable

In this study, the primary exposure variable was physical activity, assessed through a self-reported questionnaire. However, Objective tools such as accelerometer were not employed, which may introduce potential bias due to inaccurate recall or reporting. Physical activity was categorized into three distinct types: (1) High-intensity physical activity, including activities such as carrying heavy loads, digging, farming, aerobics, fast biking, cycling, and unloading heavy loads; (2) Moderate-intensity physical activity, encompasses activities such as carrying light loads, biking at a regular speed, mopping the floor, practicing tai chi, and brisk walking; (3) Low-intensity physical activity, including activities such as walking—whether for transportation (e.g., walking from one place to another at work or at home) or for leisure, sport, exercise, or recreation.

The duration of physical activity was classified into five categories based on the CHARLS questionnaire: 0 minutes, 10–29 minutes, 30–119 minutes, 120–239 minutes, and ≥240 minutes. The median value of each category was used to calculate the duration of high-, moderate-, and low-intensity physical activity [17]. The weekly duration of physical activity was calculated by multiplying the number of days the activity was performed by the daily duration of the activity.

The metabolic equivalent (MET) was used to assess the intensity of various physical activities. According to the International Physical Activity Questionnaire (IPAQ), the MET values for low-, moderate-, and high-intensity physical activity are 3.3, 4.0, and 8.0 METs, respectively [18].

Weekly PAL were calculated by multiplying the total time spent in high-intensity physical activity by 8.0, the total time spent in moderate-intensity physical activity by 4.0, and the total time spent walking by 3.3 [19].

Participants were categorized based on their weekly PALs into three groups: low (<600 MET-minutes per week), moderate (600−3,000 MET-minutes per week) or high (>3,000 MET-minutes per week) [20,21].

The principal outcome variable of interest in this study was CVD. Data on CVD were obtained from the CHARLS database using self-reported questionnaires. In the CHARLS data set, the diagnostic criteria for CVD were based on self-reported physician-diagnosed heart disease (including heart attack, coronary heart disease, angina, congestive heart failure, and other heart problems) or stroke [22].

## 2.3 Covariates assessment

Consistent with previous research, this study incorporated relevant covariates, including sociodemographic and lifestyle factors. Sociodemographic characteristics included age (≥45 years), gender (male, female), education level (no formal education, primary education, secondary education and above), residence (city, rural), and marital status (with spouse, without spouse). Data on respondents' annual household consumption were treated as a continuous variable. Lifestyle characteristics encompassed social activity and depressive symptoms. Social activity was classified based on the self-reported response to the question, "Did you engage in any of the following social activities in the past month?" Participants who selected "None of these" were classified as having no social activities, while all others were categorized as having social activities. Depression symptoms were assessed using the Centre for Epidemiologic Studies Depression Scale (CES-D), revised by Andresen in 1994. The scale consists of 10 items designed to measure the frequency of depressive symptoms experienced by the respondent during the past week. The total score ranges from 0 to 30. Based on study-defined criteria, individuals with a score below 10 were classified as not exhibiting depressive symptoms, while a score of 10 or higher indicated the presence of depressive symptoms [23]. The diagnostic criteria for hypertension, dyslipidemia, and diabetes mellitus were based on self-reported physician diagnoses. Smoking status was categorized using CHARLS data, in which participants were asked, "Have you ever chewed tobacco, smoked a pipe, smoked self-rolled cigarettes, or smoked cigarettes/cigars?" and "Do you still have a smoking habit, or have you completely stopped smoking?" Based on these two questions, all adults were classified into three mutually exclusive groups: never smoker, current smoker, and former smoker [24]. Similarly, drinking status was categorized as either non-drinker or current drinker. Socioeconomic status was comprehensively assessed through two key dimensions: educational attainment and per capita household

expenditure [25,26]. First, participants' highest educational level was quantified using standardized values (illiterate/semiliterate = 0, sishu/elementary school = 6, middle school = 9, high school/vocational school = 12, college = 16, postgraduate = 19). Second, per capita expenditure was calculated by dividing total household expenditure by the number of household members. To account for scale differences, both education level and per capita expenditure were standardized using Z-scores, and these standardized values were summed to create a composite SES score, with higher values indicating higher SES status. Participants were then stratified into three groups using tertile categorization: low SES (bottom tertile), medium SES (middle tertile), and high SES (top tertile), ensuring balanced group sizes.

### 2.4  Statistical analysis

The data were analyzed using Stata 16.0 software (Stata Corp., USA). Continuous variables were presented as mean ± standard deviation (SD), while categorical variables were presented as percentages (%). Mean differences in quantitative variables were analyzed using analysis of variance (ANOVA) and t-test, while group differences for categorical variables were analyzed using chi-square test. Logistic regression analysis was used to calculate OR and their corresponding 95% confidence intervals (CIs), providing insight into the relationship between CVD and PALs.

In the extended analysis, multiple models were developed to evaluate the relationship between CVD and PALs, adjusting for various covariates. Model 1 presented unadjusted results. Model 2 included adjustments for baseline sociodemographic characteristics, while Model 3 incorporated additional adjustments for lifestyle characteristics and chronic disease. Restricted cubic spline (RCS) analysis was used to examine the dose-response relationship between CVD and PALs (R Project for Statistical Computing, version 4.2.3). The threshold for statistical significance was set at $P < 0.05$.

## 3.  Results

### 3.1  Participants characteristics

A total of 9,015 participants were included in the final analysis, with a mean age (±SD) of 61.21 ± 9.52 years, and 4,597(50.99%) were female. Table 1 presents the baseline characteristics of participants, categorized into CVD and non-CVD groups. Among these participants, 1,069 (11.86%) were identified as having CVD. No statistically significant differences were observed in education level, annual household consumption, or social activities between the two groups (all $P > 0.05$). However, significant differences were found in age, gender, residence, marital status, depression, hypertension, dyslipidemia, diabetes, drinking status, smoking status, socioeconomic status and PALs (all $P < 0.05$). Furthermore, Table 2 presents the characteristics of participants across PALs, with 1,236 (13.71%) classified as low PALs, 2,195 (24.35%) as moderate PALs, and 5,584 (61.94%) as high PALs. The prevalence of CVD was 16.5% in the low PALs group, 13.76% in the moderate PALs group, and 10.08% in the high PALs group. Significant differences were observed among the three groups in age, education level, residence, marital status, social activities, depression, hypertension, dyslipidemia, diabetes, drinking status, smoking status, socioeconomic status and CVD (all $P < 0.05$).

### 3.2  Relationship between PALs and CVD

As shown in Table 3, in all three models, moderate and high PALs were significantly and consistently associated with a lower risk of CVD. [Crude Model: Moderate PALs: OR=0.81,95% CI: 0.67–0.98, $P < 0.05$, High PALs: OR=0.57,95%CI: 0.48–0.67, $P < 0.001$; Model 1: Moderate PALs: OR=0.80,95% CI: 0.66–0.98, $P < 0.05$ High PALs: OR=0.65, 95% CI: 0.55–0.78, $P < 0.001$; Model 2: Moderate PALs: OR=0.79,95% CI: 0.64–0.97, $P < 0.05$, High PALs: OR=0.72, 95% CI: 0.60–0.87, $P < 0.05$]. In addition, the results for physical activity quartile showed that higher levels of physical activity were consistently associated with a lower risk of CVD prevalence. This trend was consistent across the three categories, with the most significant effect observed in the highest PALs group. [Crude Model: OR=0.43,95% CI: 0.35–0.52, $P < 0.001$; Model 1: OR=0.54,95% CI: 0.44–0.67, $P < 0.001$; Model 2: OR=0.64,95% CI: 0.520.79, $P < 0.001$]. The results remained robust across all models, even after adjusting for potential confounders.

**Table 1. Fundamental Characteristics of Participants with and without CVD.**

| Characteristics | Total (N = 9015) | Non-CVD (N = 7946) | CVD (N = 1069) | χ2/ t | P value |
|---|---|---|---|---|---|
| Age (years, M ± SD) | 61.21 ± 9.52 | 60.83 ± 9.49 | 64.00 ± 9.27 | −10.28 | <.001*** |
| Gender | | | | 6.096 | .014* |
| Male | 4418(49.01) | 3932(49.48) | 486(45.46) | | |
| Female | 4597(50.99) | 4014(50.52) | 583(54.54) | | |
| Education | | | | 2.229 | .328 |
| No formal education | 1867(20.71) | 1628(20.49) | 239(22.36) | | |
| Primary education | 4163(46.18) | 3686(46.39) | 477(44.62) | | |
| Secondary education and above | 2985(33.11) | 2632(33.12) | 353(33.02) | | |
| Residence | | | | 36.927 | <.001*** |
| City | 2392(26.53) | 2026(25.50) | 366(34.24) | | |
| Rural | 6623(73.47) | 5920(74.50) | 703(65.76) | | |
| Marital status | | | | 4.912 | .027* |
| With spouse | 7755(86.02) | 6859(86.32) | 896(83.82) | | |
| Without spouse | 1260(13.98) | 1087(13.68) | 173(16.18) | | |
| Annual household consumption (M ± SD) | 10.00 ± 1.20 | 10.00 ± 1.19 | 9.94 ± 1.16 | 1.594 | .11 |
| Social Activities | | | | 0.632 | .427 |
| With Social activities | 4797(53.21) | 4216(53.06) | 581(54.35) | | |
| No social activities | 4218(46.79) | 3730(46.94) | 488(45.65) | | |
| Depression | | | | 70.647 | <.001*** |
| Suffer | 3598(39.91) | 3045(38.32) | 553(51.73) | | |
| Not suffering | 5417(60.09) | 4901(61.68) | 516(48.27) | | |
| Hypertension | | | | 312.739 | <.001*** |
| Hypertension | 3104(34.43) | 2478(31.19) | 626(58.56) | | |
| Non-hypertension | 5911(65.57) | 5468(68.81) | 443(41.44) | | |
| Dyslipidemia | | | | 320.448 | <.001*** |
| Dyslipidemia | 1667(18.49) | 1256(15.81) | 411(38.45) | | |
| Non-Dyslipidemia | 7348(81.51) | 6690(84.19) | 658(61.55) | | |
| Diabetes | | | | 129.634 | <.001*** |
| Diabetes | 1010(11.20) | 780(9.82) | 230(21.52) | | |
| Non-Diabetes | 8005(88.80) | 7166(90.18) | 839(78.48) | | |
| Drinking status | | | | 27.991 | <.001*** |
| Non-drinker | 3158(35.03) | 2861(36.01) | 297(27.78) | | |
| Current drinker | 5857(64.97) | 5085(63.99) | 772(72.22) | | |
| Smoking status | | | | 16.453 | <.001*** |
| Non-smoker | 5145(57.07) | 4517(56.85) | 628(58.75) | | |
| Current smoker | 1007(11.17) | 858(10.80) | 149(13.94) | | |
| Former smoker | 2863(31.76) | 2571(32.36) | 292(27.32) | | |
| Socioeconomic status | | | | 16.858 | <.001*** |
| Bottom tertile | 3000(33.28) | 2612(32.87) | 388(36.30) | | |
| Middle tertile | 3005(33.33) | 2708(34.08) | 297(27.78) | | |
| Top tertile | 3010(33.39) | 2626(33.05) | 384(35.92) | | |
| PAL | | | | 49.965 | <.001*** |
| Low PAL | 1236(13.71) | 1032(12.99) | 204(19.08) | | |

*(Continued)*

**Table 1.** (Continued)

| Characteristics | Total (N = 9015) | Non-CVD (N = 7946) | CVD (N = 1069) | χ2/ t | P value |
|---|---|---|---|---|---|
| Moderate PAL | 2195(24.35) | 1893(23.82) | 302(28.25) | | |
| High PAL | 5584(61.94) | 5021(63.19) | 563(52.67) | | |

M±SD = mean ± standard deviation,

*P<.05,

**P<.01,

***P<.001.

Subsequent analysis using a restricted cubic spline model based on Model 2 revealed a significant inverse linear relationship between PALs and the risk of prevalent CVD (non-linearity P = 0.398). Notably, when PALs reached 4,068 MET-minutes/week, the OR was observed to be < 1. As illustrated in Fig 2, the overall trend of the curve demonstrates a negative correlation, with OR values decreasing as amount of physical activity increase. This indicates that higher levels of physical activity were associated with a lower risk of CVD prevalence. At weekly physical activity levels exceeding 10,000 MET-minutes/week, the curve begins to plateau, suggesting that the additional health benefits of further increases in physical activity may diminish. Furthermore, the confidence intervals were narrow at moderate and high physical activity levels, indicating greater statistical stability. However, at extreme MET-minutes/week values (<600 or >12,000), the confidence intervals widened, reflecting increased uncertainty due to fewer data points within these ranges.

## 4. Discussion

CVD represents a significant global health concern, encompassing a wide range of circulatory disorders, including heart and blood vessel diseases, pulmonary circulatory diseases, and cerebrovascular diseases [27]. Currently, CVD was the leading cause of death among both urban and rural residents [28]. Moreover, the burden of CVD continues to grow, posing a major public health challenge [29]. It was imperative for governments to take the lead in preventing and controlling CVD [30]. CVD was driven by multiple risk factors, making its prevention and control inherently complex and requiring a multifaceted approach [31]. A substantial body of epidemiological evidence had identified the primary CVD risk factors in China as physical inactivity, hypertension, dyslipidemia, smoking, and overweight or obesity [32–34]. This study aims to examine the association between PAL and CVD in a nationally representative sample of middle-aged and older adults in China. Understanding this relationship was critical for identifying high-risk populations and developing targeted interventions and prevention strategies.

Our results showed that the prevalence of CVD in the middle-aged and older adult population was 11.86%, which was similar to the findings from the previous studies [35]. However, a study by Zhang Z Y et al. [36], found that the prevalence of CVD in China increase by 4.22% from 1990 to 2019, rising from 4.24% to 8.46%, which was lower than the prevalence observed in this study. Cheng J et al. [37], reported that the overall prevalence of CVD in Shanghai continued to rise from 2016 to 2020, increase from 17.6% in 2016 to 19.6% in 2020, which was higher than the prevalence in our study. These differences in prevalence may be attributed to variations in the regions and age groups studied. In our study, the prevalence of CVD was also influenced by multiple factors, including age, gender, geographic location, marital status, social engagement, depressive disorders, and PALs, which aligns with findings from previous studies [38–43]. Furthermore, hypertension, diabetes, and dyslipidemia were recognized as common risk factors for cardiovascular disease [44]. Behavioral factors such as smoking and excessive alcohol consumption may exacerbate vascular dysfunction by increasing oxidative stress and chronic inflammation [45,46]. By accounting for these covariates, our model enhances the robustness of the observed relationship between physical activity and cardiovascular outcomes.

**Table 2. Fundamental Characteristics of Participants by PAL.**

| Characteristics | Total population (N = 9015) | Low PAL (N = 1236) | Moderate PAL (N = 2195) | High PAL (N = 5584) | χ2/ F | *P* value |
|---|---|---|---|---|---|---|
| Age (years, M ± SD) | 61.21 ± 9.52 | 64.48 ± 10.83 | 62.32 ± 9.84 | 60.04 ± 8.83 | 7.596 | <.001*** |
| Gender | | | | | 2.043 | .360 |
| Male | 4418 (49.01) | 629 (50.89) | 1067 (48.61) | 2722 (48.75) | | |
| Female | 4597 (50.99) | 607 (49.11) | 1128 (51.39) | 2862 (51.25) | | |
| Education | | | | | 71.753 | <.001*** |
| No formal education | 1867 (20.71) | 315 (25.49) | 372 (16.94) | 1180 (21.13) | | |
| Primary education | 4163 (46.18) | 578 (46.76) | 960 (43.74) | 2625( 47.01) | | |
| Secondary education and above | 2985 (33.11) | 343 (27.75) | 863 (39.32) | 1779 (31.86) | | |
| Residence | | | | | 213.394 | <.001*** |
| City | 2392 (26.53) | 289 (23.38) | 845 (38.50) | 1258 (22.53) | | |
| Rural | 6623 (73.47) | 947 (76.62) | 1350 (61.50) | 4326 (77.47) | | |
| Marital status | | | | | 70.795 | <.001*** |
| With spouse | 7755 (86.02) | 1005 (81.31) | 1813 (82.60) | 4937 (88.41) | | |
| Without spouse | 1260 (13.98) | 231 (18.69) | 382 (17.40) | 647 (11.59) | | |
| Annual household consumption | 10.00 ± 1.20 | 9.98 ± 1.22 | 9.99 ± 1.17 | 10.00 ± 1.19 | .959 | .909 |
| Social Activities | | | | | 87.078 | <.001*** |
| With Social Activities | 4797 (53.21) | 514 (41.59) | 1269 (57.81) | 3014 (53.98) | | |
| No Social Activities | 4218 (46.79) | 722 (58.41) | 926 (42.19) | 2570 (46.02) | | |
| Depression | | | | | 31.343 | <.001*** |
| Suffer | 3598 (39.91) | 576 (46.60) | 811 (36.95) | 2211 (39.60) | | |
| Not Suffering | 5417 (60.09) | 660 (53.40) | 1384 (63.05) | 3373 (60.40) | | |
| Hypertension | | | | | 61.679 | <.001*** |
| Hypertension | 3104 (34.43) | 507 (41.02) | 843 (38.41) | 1754 (31.41) | | |
| Non-hypertension | 5911 (65.57) | 729 (58.98) | 1352 (61.59) | 3830 (68.59) | | |
| Dyslipidemia | | | | | 39.789 | <.001*** |
| Dyslipidemia | 1667 (18.49) | 225 (18.20) | 504 (22.96) | 938 (16.80) | | |
| Non-Dyslipidemia | 7348 (81.51) | 1011 (81.80) | 1691 (77.04) | 4646 (83.20) | | |
| Diabetes | | | | | 29.683 | <.001*** |

*(Continued)*

**Table 2.** (Continued)

| Characteristics | Total population (N = 9015) | Low PAL (N = 1236) | Moderate PAL (N = 2195) | High PAL (N = 5584) | χ2/ F | P value |
|---|---|---|---|---|---|---|
| Diabetes | 1010 (11.20) | 142 (11.49) | 313 (14.26) | 555 (9.94) | | |
| Non-Diabetes | 8005 (88.80) | 1094 (88.51) | 1882 (85.74) | 5029 (90.06) | | |
| Drinking status | | | | | 29.017 | <.001*** |
| Non-drinker | 3158 (35.03) | 356 (28.80) | 750 (34.17) | 2052 (36.75) | | |
| Current drinker | 5857 (64.97) | 880 (71.20) | 1445 (65.83) | 3532 (63.25) | | |
| Smoking status | | | | | 17.988 | .001** |
| Non-smoker | 5145 (57.07) | 655 (52.99) | 1286 (58.59) | 3204 (57.38) | | |
| Current smoker | 1007 (11.17) | 160 (12.94) | 263 (11.98) | 584 (10.46) | | |
| Former smoker | 2863 (31.76) | 421 (34.06) | 646 (29.43) | 1796 (32.16) | | |
| Socioeconomic status | | | | | 71.897 | <.001*** |
| Bottom tertile | 3000 (33.28) | 475 (38.43) | 620 (28.24) | 1905 (34.12) | | |
| Middle tertile | 3005 (33.33) | 358 (28.96) | 708 (32.26) | 1939 (34.72) | | |
| Top tertile | 3010 (33.39) | 403 (32.61) | 867 (39.50) | 1740 (31.16) | | |
| CVD | | | | | 49.965 | <.001*** |
| Yes | 1069 (11.86) | 204 (16.50) | 302 (13.76) | 563 (10.08) | | |
| No | 7946 (88.14) | 1032 (83.50) | 1893 (86.24) | 5021 (89.92) | | |

OR = odds ratio; CI = confidence interval.

*$P$<.05,

**$P$<.01,

***$P$<.001.

This study shows that, compared to the low PAL group in Model 2, the risk of CVD was relatively lower in both the moderate PALs group (OR = 0.79, 95% CI: 0.64–0.97) and the high PALs group (OR = 0.71, 95% CI: 0.59–0.86). To verify the robustness of our primary findings, we conducted sensitivity analyses by reclassifying PALs from the original three-category grouping (low/medium/high) into quartiles (Q1-Q4). As physical activity increased from Q1 (<1,733 MET-minutes/week) to Q4 (>9,198 MET-minutes/week), the odds ratios (ORs) for CVD risk demonstrated a clear graded decline (OR decreasing from 0.90 to 0.64, P for trend <0.05). This monotonic decreasing trend further supports the protective effect of physical activity. Specifically, compared to the least active group (Q1), the Q2 group (1,733–4,158 MET-minutes/week) showed a 10% reduction in risk (OR = 0.90, 95% CI: 0.75–1.07), the Q3 group (4,158–9,198 MET-minutes/week) exhibited a significant 22% decrease (OR = 0.78, 95% CI: 0.65–0.94), and the Q4 group demonstrated the greatest reduction at 36% (OR = 0.64, 95% CI: 0.52–0.79). These findings align with those of a prospective cohort study by Lear SA *et al*. [20], higher PAL may influence the risk of CVD prevalence by improving key risk factors such as blood pressure, lipid profile, blood glucose levels, and body weight [47]. Additionally, a review on the effects of physical activity on atherosclerosis

**Table 3. Logistic Regression Analysis of the Association between PALs (MET-minutes/week) and CVD.**

| Dependent variable | Crude Model | | Model 1 | | Model 2 | |
|---|---|---|---|---|---|---|
| | OR (95%CI) | P | OR (95%CI) | P | OR (95%CI) | P |
| PAL | | | | | | |
| Low PAL : <600 | 1.00 (Ref) | | 1.00 (Ref) | | 1.00 (Ref) | |
| Moderate PAL : 600~3000 | 0.81 (0.67~0.98) | .030* | 0.80 (0.66~0.98) | .028* | 0.79 (0.64~0.97) | .026* |
| High PAL :>3000 | 0.57 (0.48~0.67) | <.001** | 0.65 (0.55~0.78) | <.001** | 0.72 (0.60~0.87) | .001* |
| PAL quartile | | | | | | |
| Q1 : <1733 | 1.00 (Ref) | | 1.00 (Ref) | | 1.00 (Ref) | |
| Q2 : 1733~4158 | 0.81 (0.69~0.96) | .013* | 0.84 (0.71~0.99) | .038* | 0.90 (0.75~1.07) | .225 |
| Q3 : 4158~9198 | 0.63 (0.53~0.75) | <.001** | 0.70 (0.59~0.84) | <.001** | 0.78 (0.65~0.94) | <.008* |
| Q4 : >9198 | 0.43 (0.35~0.52) | <.001** | 0.54 (0.44~0.67) | <.001** | 0.64 (0.52~0.79) | <.001** |

In the analysis presented in Table 3, PALs were treated as a categorical variable and analyzed using logistic regression. Three models were constructed: Crude model: non-adjusted; Model 1: adjusted for age, gender, education, residence, marital status, and annual household consumption; Model 2: Model 1 plus adjustments for social activities, depression, hypertension, dyslipidemia, diabetes, drinking status, smoking status, Socioeconomic status. The associations between PALs and the risk of CVD were estimated using the logistic regression.

Abbreviations: Ref = reference; OR=odds ratio; CI = confidence interval.

*$P < .05$,

**$P < .001$.

biomarkers in CVD provided robust evidence that physical activity reduces pro-inflammatory markers, including TNF-α, IL-6, CRP, and vascular endothelial growth factor, while also decreasing vascular cell adhesion molecules. These mechanisms collectively contribute to a protective effect against CVD development [48].

In this study, 61.94% of participants (5,584 individuals) were categorized as having a high PALs, defined as exceeding 3,000 MET-minutes/week. This distribution provided a unique opportunity to examine the relationship between high PALs and CVD risk. A negative linear relationship between weekly PALs and CVD risk was analyzed using restricted cubic spline graphs. The restricted cubic spline plot revealed a significant inverse linear trend between weekly physical activity levels (ranging from 0 to 15,000 MET-minutes/week) and CVD risk. Specifically, higher levels of physical activity are consistently associated with a lower risk of CVD prevalence. The odds ratio demonstrated a linear decline as physical activity levels increased. However, the curve began to plateau when physical activity exceeded 10,000 MET-minutes/week. This phenomenon suggests the potential existence of a threshold effect in the relationship between physical activity and CVD risk. Specifically, further increases in physical activity beyond a certain level may not significantly contribute to additional reductions in CVD risk. From a biological mechanism perspective, the health benefits of physical activity typically follow a nonlinear dose-response curve [49]. Regular, moderate exercise can significantly reduce the risk of various chronic diseases by improving insulin sensitivity [50], reducing systemic inflammation [51], and enhancing cardiopulmonary function. However, when physical activity exceeds a certain threshold, the body's metabolic and organ systems reach relative saturation, potentially leading to diminishing health returns. This manifests as a plateau effect where the magnitude of risk reduction gradually stabilizes. This finding aligns with existing research, which indicates that the health benefits of physical activity may plateau beyond a certain threshold, and excessive exercise could potentially introduce other health risks [52]. A large cohort study from PURE (Prospective Urban Rural Epidemiology) demonstrated graded reductions in mortality (HR 0.80, 95% CI 0.74–0.87 and 0.65, 0.60–0.71; p-trend<0.0001) and major CVD (0.86, 0.78–0.93;

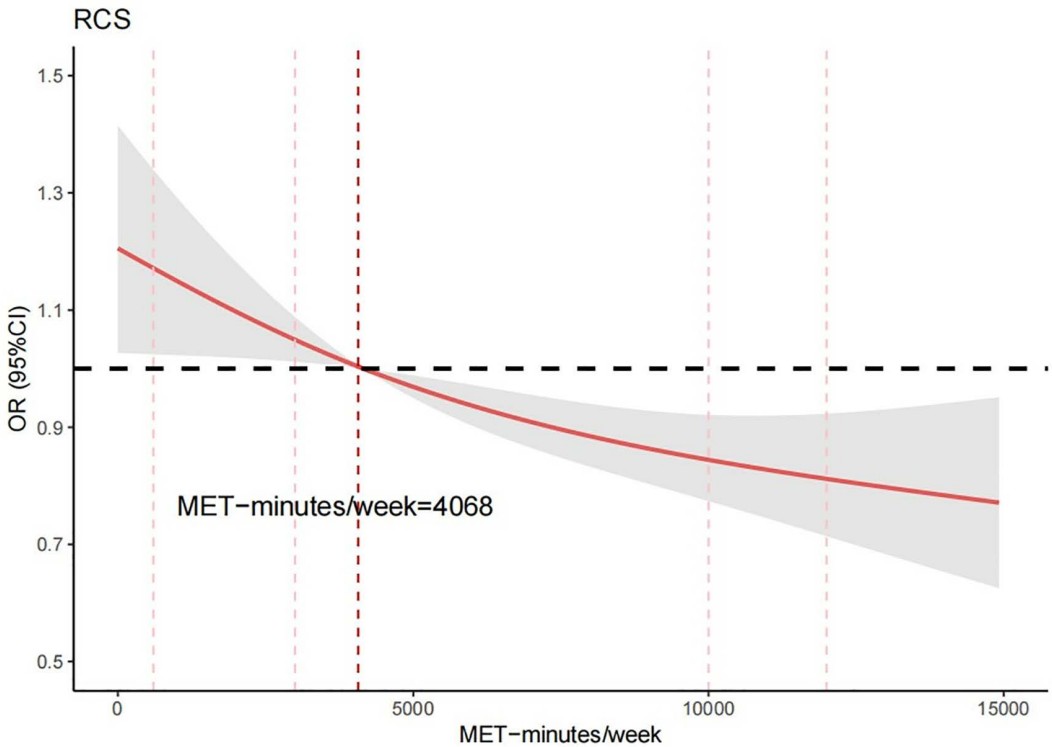

**Fig 2. Restricted cubic spline plots showing the association between changes in PAL and risk of CVD.** Association between amount of physical activity and cardiovascular disease risk. This figure presented the results of RCS analysis based on Model 2, illustrating the dose-response relationship between physical activity (measured in MET-minutes/week) and the OR for cardiovascular disease. The OR values were represented by the solid red line, with the 95% confidence intervals (95% CI) indicated by the shaded gray area. The graph was truncated at 15,000 MET-minutes/week, corresponding to the 93rd percentile of the physical activity distribution. The red vertical dashed lines indicated key thresholds at 600, 3,000, 10,000, and 12,000 MET-minutes per week. The confidence intervals were wider at low (<600 MET-minutes/week) and high (>12,000 MET-minutes/week) levels of physical activity, which may be due to smaller sample sizes or greater data variability at these extremes. In contrast, within the 3,000–10,000 MET-minutes/week range, the sample size was larger and more stable, resulting in narrower confidence intervals. Overall, the trend showed that the risk ratio decreased progressively with increasing physical activity.

p-trend<0.001) when comparing moderate (600–3000 MET-minutes/week) and high (>3000 MET-minutes/week) activity levels against low activity (<600 MET-minutes) [20]. Another large cohort study based on the UK Biobank identified a significant linear dose-response relationship between moderate, vigorous, and total PAL and the risk of incident CVD [53]. This result partially aligns with the conclusions of the present study. Additionally, the study by Bakker EA *et al.* [14], identified a dose-response relationship between PAL, major adverse cardiovascular events (MACE), and all-cause mortality across different population groups: healthy individuals, individuals with cardiovascular risk factors(CVRF), and individuals with CVD. PAL, measured in MET-minutes per week, was divided into quartiles: Q1 (1–1912 MET-minutes/week), Q2 (1913–3690 MET-minutes/week), Q3 (3691–7257 MET-minutes/week), and Q4 (>7258 MET-minutes/week). The study found that higher PAL was associated with a lower risk of adverse outcomes in both healthy individuals and patients with CVRF. For CVD patients, only the highest PA quartile was associated with a lower risk of adverse prognosis. The PAL reported in this study aligns with our findings and exceeds the World Health Organization's (WHO) recommendations for both adults and older adults [54]. This may be attributed to the inclusion of non-exercise daily physical activities. To meet the physical activity guidelines established by the Food and Agriculture Organization (FAO), WHO and United Nations University (UNU), individuals were encouraged to engage in a weekly physical activity regimen ranging from 3,000–6,000

MET-minutes per week [13]. However, extremely high activity levels may not be absolutely safe. Research indicates that sustained excessive exercise may trigger overtraining syndrome, potentially inducing learning and memory impairments through elevated inflammatory cytokines and oxidative stress markers [55]. Additionally, studies report that chronic endurance athletes may develop atrial dilation and increased arrhythmia risk. In summary, physical activity exceeding 10,000 MET-minutes/week may reach a "physiological saturation point" – while not immediately hazardous, the health benefits plateau. Beyond this threshold, certain individuals may experience diminishing returns or potential adverse effects with further training load escalation.

Our findings suggest a significant association between PALs and CVD; however, the possibility of reverse causality must be considered. Individuals with CVD may reduce their physical activity levels due to disease-related limitations [56], leading to an overestimation of the observed association. Future research should employ longitudinal designs to better elucidate the causal pathways linking PALs and CVD risk. In addition to reverse causality, socioeconomic status (SES) should also be considered, as individuals with lower SES face greater barriers to accessing healthcare resources and engaging in physical activity. Those with lower SES may struggle to obtain adequate healthcare services or maintain sufficient physical activity levels due to economic constraints [57]. A study found that participants with low SES had a 1.65 to 2.25 times higher risk of developing CVD compared to those with high SES, with lifestyle factors mediating only 3% to 12% of this association [58]. This suggested that the health risks among individuals with low SES were not solely attributable to unhealthy lifestyles but were also influenced by inequalities in healthcare access. Research had shown a significant association between education level and household consumption expenditure [59], and both metrics reflect different dimensions of an individual's socioeconomic status [60,61].

The findings of this study were based on a cohort of middle-aged and elderly individuals in China, and it was important to consider the extent to which these results can be generalized to other populations. However, notable differences existed in physical activity patterns between China and Western countries. In Western countries, particularly in Europe and North America, structured forms of exercise-such as gym-based activities, aerobic classes, and organized fitness programs-were more prevalent. These activities were typically planned, organized, and primarily aimed at improving health and physical fitness. As highlighted by Hallal PC et al. [62], physical activity in Western populations was more closely associated with structured exercise. In contrast, in China, physical activity was predominantly linked to daily living activities and occupational labor rather than structured exercise. These differences may influence the observed associations between PAL and CVD risk. Although the dose-response relationship observed in this study aligned with findings from Western populations [20], caution should be exercised when generalizing these results to younger individuals, different ethnic groups, or non-Asian populations. Furthermore, research by Li J et al. [63], had demonstrated significant differences in occupational physical activity (OPA) patterns between urban and rural areas in China. In rural areas, participants were more likely to engage in labor-intensive work, such as agricultural labor, which required substantial physical exertion. In contrast, urban participants were more often employed in office-based or other low physical activity occupations. Rural adults exhibited higher levels of occupational moderate-to-vigorous physical activity (MVPA), which was significantly associated with lower risks of all-cause and CVD mortality. Conversely, urban areas showed lower levels of occupational MVPA, and no significant association with mortality was observed. These findings underscored the importance of urban-rural differences in the relationship between OPA and health outcomes. In our cohort, rural residents accounted for 73.47% of participants compared to 26.53% urban dwellers. Consequently, the applicability of our findings may be limited in populations where leisure-time physical activity serves as the primary form of exercise.

This study has several limitations. First, its cross-sectional design limits the ability to establish causality; future research incorporating prospective longitudinal data would provide stronger insights. Second, although we adjusted for several key confounders, the possibility of residual confounding due to unmeasured variables or healthier lifestyle behaviors associated with higher PAL cannot be excluded. Future studies should incorporate additional potential confounding factors, such as body mass index (BMI), high-density lipoprotein cholesterol (HDL-C), low-density lipoprotein cholesterol (LDL-C),

triglycerides (TG) and diet, medication adherence, genetic factors, into the analysis to better elucidate these relationships. Third, consistent with previous studies [16,64,65], the determination of CVD in this study was based on self-reported physician diagnoses, which may be subject to recall bias and misclassification bias, particularly among participants with subjective symptoms but no formal diagnosis. Fourth, the use of list wise deletion for missing data excluded over 10,000 participants from the original data set. As missingness correlated with baseline characteristics, selection bias cannot be dismissed, potentially affecting the representativeness and generalizability of our findings.

Despite these limitations, our study had several notable strengths. First, we utilized a comprehensive data resource from the largest current population in China addressing the health and social transition of the elderly population, which was more reliable than studies based on hospital data. Second, the data for our study were obtained through face-to-face interviews, which were more reliable than telephone interviews. Third, the method used to assess PALs was a significant advantage. Given the variety of methods employed to assess PALs and the absence of consensus on criteria for defining PALs, this study utilized a questionnaire (IPAQ version) that has been validated in multiple studies and across diverse populations [66,67]. The questionnaire inquired about the intensity, frequency, and duration of activity, enabling the investigation of associations between physical activity and CVD. This approach helped to minimize the risk of misclassification, overestimation, and underestimation. Although prior studies indicate a relatively low misclassification rate for self-reported CVD, these findings should be interpreted with caution given the substantial proportion of rural participants in the CHARLS cohort in China, along with their limited CVD awareness and potential recall bias, which may increase misclassification risk.

## 5. Conclusion

In summary, our study reveals a negative correlation between PAL and the prevalence of CVD. Furthermore, our findings suggest that middle-aged and older adults should maintain moderate to high levels of physical activity, as this is associated with a lower risk of CVD.

## Supporting information

**S1 File. STROBE-checklist-v4-combined-PlosMedicine.**
(DOCX)

**S2 File. Data.**
(XLS)

## Acknowledgments

We would like to express our gratitude to the National Development Research Institute of Peking University and the Peking University China Social Science Research Centre for providing the CHARLS data.

## Author contributions

**Conceptualization:** Haiwei Li, Liang Luo, Jing Zhang, Yanhao Zhao, Peipei Cheng, Dan Liu, Liwei Guo.

**Formal analysis:** Haiwei Li, Liang Luo.

**Investigation:** Haiwei Li, Liang Luo, Jing Zhang, Peipei Cheng.

**Methodology:** Haiwei Li, Liang Luo, Yanhao Zhao.

**Resources:** Liang Luo, Jing Zhang, Yanhao Zhao.

**Supervision:** Yanhao Zhao, Peipei Cheng, Dan Liu.

**Visualization:** Jing Zhang.

**Writing – original draft:** Haiwei Li, Liang Luo, Jing Zhang.

**Writing – review & editing:** Haiwei Li, Liang Luo, Jing Zhang, Peipei Cheng, Dan Liu, Liwei Guo.

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
