## [Decision Letter · Decision Letter 0]

PONE-D-24-53272Association between physical activity level and cardiovascular disease: an empirical analysis based on CHARLS data in 2018PLOS ONE

Dear Dr. Li,

Thank you for submitting your manuscript to PLOS ONE. After careful consideration, we feel that this study has merits and it really adds to the existing literature, but needs some major revisions before publication. Therefore, we invite you to submit a revised version of the manuscript that addresses the points raised during the review process.

Please kindly address the comments raised by our peer reviewers, especially on following aspects:1. Statistical anlayis: Adjusting for key confounders2. Explanation on generalizability of your findings3. Proper referencing 4. Use of standard and intelligible language in the manuscript 

We look forward to receiving your revised manuscript.

Kind regards,

Bishnu Deep Pathak, MBBS

Academic Editor

PLOS ONE

“The study was financially supported by the Shanxi Province Philosophy and Social Science Planning Project (NO. 2020YY080). The funder had no role in the study design, data collection and analysis, decision to publish, or preparation of the manuscript.”

“HL was supported by the Shanxi Province Philosophy and Social Science Planning Project (No. 2020YY080). The funder had no role in the study design, data collection and analysis, decision to publish, or preparation of the manuscript.”

3. We note that there is identifying data in the Supporting Information file < S2 File-data.xls>. Due to the inclusion of these potentially identifying data, we have removed this file from your file inventory. Prior to sharing human research participant data, authors should consult with an ethics committee to ensure data are shared in accordance with participant consent and all applicable local laws.

-Location data

Reviewers' comments:

Reviewer's Responses to Questions

**Comments to the Author**

1. Is the manuscript technically sound, and do the data support the conclusions?

Reviewer #1: Yes

Reviewer #2: Yes

2. Has the statistical analysis been performed appropriately and rigorously? 

Reviewer #1: Yes

Reviewer #2: Yes

3. Have the authors made all data underlying the findings in their manuscript fully available?

Reviewer #1: Yes

Reviewer #2: Yes

4. Is the manuscript presented in an intelligible fashion and written in standard English?

Reviewer #1: Yes

Reviewer #2: No

5. Review Comments to the Author

Reviewer #1: I read the manuscript titled “Association between physical activity level and cardiovascular disease: an empirical analysis based on CHARLS data in 2018” by Haiwei Li and colleagues with special interest.

This study reported significant correlation between prevalence of CVD and physical activity level. I congratulate authors for reporting their significant findings. I don’t have major comments however I have small comment in regards to the manuscript.

The referencing is not per the PLOS style and ref 23 authors name is somehow not correctly listed therefore would recommend to revisit the authors guideline (https://journals.plos.org/plosone/s/submission-guidelines) and proof read them as mentioned in the guideline “ In the text, cite the reference number in square brackets (e.g., “We used the techniques developed by our colleagues [19] to analyze the data”)”.

Reviewer #2: Feedback for Authors

This study investigates the association between physical activity levels (PAL) and cardiovascular disease (CVD) among middle-aged and elderly individuals in China using data from the 2018 China Health and Retirement Longitudinal Study (CHARLS). The study presents a valuable epidemiological analysis, highlighting a significant inverse relationship between PAL and CVD prevalence. The use of a large, nationally representative dataset strengthens its relevance, and the application of multivariate logistic regression and restricted cubic spline models enhances its methodological rigor.

However, several major revisions are required to improve the robustness, clarity, and interpretability of the findings. The main concerns revolve around (1) methodological limitations, (2) confounding adjustments, (3) self-reported CVD diagnosis, (4) PAL categorization, and (5) discussion of causality and generalizability. Addressing these issues will enhance the validity and impact of the study.

General Comments

- Methodological Limitations: Cross-sectional Design & Causality

The study employs a cross-sectional design, yet much of the discussion implies a causal relationship between PAL and CVD risk reduction. The wording throughout the manuscript should be tempered to avoid causal inferences. Statements such as “higher PAL reduces CVD risk” should be modified to indicate an association rather than causality.

A clearer acknowledgment of reverse causality is necessary. Individuals with CVD may engage in less physical activity due to disease-related limitations, leading to an overestimation of the association. The discussion should explicitly recognize this issue and suggest prospective studies to confirm the findings.

- Confounding Factors and Model Adjustments

The multivariate analysis lacks adjustment for key confounders, including hypertension, diabetes, hyperlipidemia, smoking, alcohol consumption, and BMI—all of which are strong predictors of CVD and could mediate the association with PAL.

Without adjusting for these critical variables, it remains unclear whether the observed association is due to PAL itself or reflects a clustering of other healthier lifestyle behaviors in the high PAL group. The authors should include additional confounders in the regression models or justify why these variables were excluded.

The impact of social and economic status (which may influence both PAL and access to healthcare) should be discussed more explicitly as a potential source of residual confounding.

- Self-Reported CVD Diagnosis: Accuracy & Bias

The diagnosis of CVD is based on self-reported physician-diagnosed heart disease or stroke. This method is prone to recall bias and may lead to misclassification errors. Participants with undiagnosed CVD or those with subjective symptoms but no formal diagnosis may be misclassified.

The sensitivity and specificity of self-reported CVD data should be addressed in the discussion, and comparisons with objectively measured CVD prevalence in similar populations should be provided if available.

- PAL Categorization and MET-Minute Thresholds

The classification of low, moderate, and high PAL is based on arbitrary MET-minute cutoffs (<600, 600-3000, >3000 MET-min/week). However, it is unclear whether these categories align with established guidelines such as WHO, ACSM, or AHA recommendations.

Justification should be provided for the chosen MET cutoffs. Alternatively, the analysis could be repeated using quartiles or deciles of PAL to explore whether different thresholds yield different results.

The interpretation of 3000 MET-min/week as a threshold for reduced CVD prevalence should be critically examined, as it does not align directly with WHO recommendations for physical activity.

- Restricted Cubic Spline Interpretation

The restricted cubic spline analysis suggests a linear inverse association between PAL and CVD (p > 0.05 for non-linearity). However, Figure 2 should be more clearly annotated to indicate where the confidence intervals widen, which could suggest statistical uncertainty at extreme PAL levels.

The authors should discuss whether this linear trend holds across all PAL levels or whether the association plateaus at higher MET levels, as some studies suggest diminishing returns beyond a certain point.

- Generalizability & Population-Specific Considerations

The study is based on Chinese middle-aged and elderly individuals, and lifestyle patterns may differ significantly from Western populations. The discussion should explicitly address the external validity of the findings and whether they can be generalized to younger individuals, different ethnic groups, or non-Asian populations.

Urban-rural differences in occupational physical activity should also be discussed, as many rural participants may engage in high PAL due to labor-intensive work rather than structured exercise.

Detailed Comments

Abstract

The phrase “high PAL reduces CVD risk” should be changed to “high PAL is associated with a lower CVD prevalence.”

The sample size (9015 participants) should be explicitly stated in the methods section of the abstract.

Introduction

The introduction effectively highlights the importance of PAL, but it oversimplifies the causal relationship. Statements such as "maintaining >3000 MET-min/week reduces CVD risk" should be softened to "is associated with lower CVD prevalence."

Methods

Multivariable model adjustments: Include key covariates such as hypertension, diabetes, smoking, alcohol consumption, and BMI.

Clarify PAL measurement: Was PAL derived solely from self-reported surveys, or were objective measurements (e.g., accelerometers) available?

Describe how missing data were handled: Was multiple imputation used, or were missing cases excluded?

Results

Table 1: Add p-values for baseline characteristics to indicate whether significant differences exist between groups.

Table 3: Clarify whether PAL was modeled as a categorical or continuous variable in the logistic regression.

Discussion

Causality: Strengthen the acknowledgment of reverse causality and residual confounding.

Confounding bias: Explicitly discuss the lack of adjustment for key clinical variables such as hypertension, smoking, and diabetes.

Dose-response relationship: Discuss whether there is a plateau effect for high PAL levels.

Generalizability: Address whether findings are applicable to younger or non-Chinese populations.

Figures & Tables

Figure 2: Add confidence interval shading to highlight the uncertainty at extreme PAL levels.

Table 3: Consider including effect sizes (ORs) for additional quartiles or deciles of PAL instead of only three broad categories.

6. PLOS authors have the option to publish the peer review history of their article (what does this mean? ). If published, this will include your full peer review and any attached files.

**Do you want your identity to be public for this peer review?** For information about this choice, including consent withdrawal, please see our Privacy Policy .

Reviewer #1: No

Reviewer #2: No

---

## [Author Response · Author response to Decision Letter 1]

29 Mar 2025

Response to reviewers

We are so grateful for the thoughtful reviews provided by the external referees. We have tried to carefully address each recommendation. We believe that our manuscript is now stronger as a result of the modifications outlined below. We also thank the editoral staff and peer-reviewers for their contributions.

The modified text is highlighted in yellow in the revised manuscript. In addition to the specific responses to the referees shown below, we performed substantial language revising according reviews and editoral staff advise.

Reviewer #1

The referencing is not per the PLOS style and ref 23 authors name is somehow not correctly listed therefore would recommend to revisit the authors guideline (https://journals.plos.org/plosone/s/submission-guidelines) and proof read them as mentioned in the guideline ; In the text, cite the reference number in square brackets (We used the techniques developed by our colleagues [19] to analyze the data)

Response: We have carefully reviewed the PLOS ONE author guidelines and revised the references to ensure they fully comply with the journal’s style. Specifically, we have:

1. Corrected the formatting of all references to align with PLOS ONE requirements.

2. Verified and updated the author names in reference 23 to ensure accuracy.

3. Ensured that in-text citations are consistently formatted with square brackets, as per the guidelines (e.g., [19]).

Reference See line�410-611

Reviewer #2

General Comments

(1) - Methodological Limitations: Cross-sectional Design and Causality

The study employs a cross-sectional design, yet much of the discussion implies a causal relationship between PAL and CVD risk reduction. The wording throughout the manuscript should be tempered to avoid causal inferences. Statements such as; higher PAL reduces CVD risk; should be modified to indicate an association rather than causality. A clearer acknowledgment of reverse causality is necessary. Individuals with CVD may engage in less physical activity due to disease-related limitations, leading to an overestimation of the association. The discussion should explicitly recognize this issue and suggest prospective studies to confirm the findings.

Response:

We have thoroughly reviewed the manuscript and modified all statements that may imply causality. For example, "higher PAL reduces the risk of adverse outcomes" has been revised to "higher PAL is associated with a lower risk of adverse outcomes." See line�299-302

In response to your suggestions, a discussion on the reverse relationship between physical activity and cardiovascular disease has been added to the discussion section. It is also recommended that future studies adopt a prospective design to better elucidate the association between PAL and CVD risk. For discussion additions, See line�316-320

(2)- Confounding Factors and Model Adjustments

The multivariate analysis lacks adjustment for key confounders, including hypertension, diabetes, hyperlipidemia, smoking, alcohol consumption, and BMI; all of which are strong predictors of CVD and could mediate the association with PAL.

Without adjusting for these critical variables, it remains unclear whether the observed association is due to PAL itself or reflects a clustering of other healthier lifestyle behaviors in the high PAL group. The authors should include additional confounders in the regression models or justify why these variables were excluded. The impact of social and economic status (which may influence both PAL and access to healthcare) should be discussed more explicitly as a potential source of residual confounding.

Response:

In this study, we re-analyzed the multivariate regression model by incorporating the covariates hypertension, dyslipidemia, diabetes, drinking status, smoking status. These variables were available in CHARLS dataset, and their inclusion provided a more comprehensive perspective for assessing the relationship between PAL and CVD risk. Additionally, we considered the inclusion of potential confounding factors such as BMI, high-density lipoprotein cholesterol (HDL-C), low-density lipoprotein cholesterol (LDL-C), and triglycerides (TG) to further control for confounding effects. However, due to the lack of physical examination and blood test data in the 2018 CHARLS dataset, these variables were ultimately not included in the analysis. In the limitations, we recommend that future studies incorporate these variables to enable a more comprehensive evaluation of the relationship between PAL and CVD risk. For further details on the CHARLS database, please refer to its official website : https://charls.pku.edu.cn/. For newly added variables, See line : 221. For limitations additions, See line�356-361.

Furthermore, although this study did not include socioeconomic status (SES) as a separate covariate in the analysis, we incorporated both educational attainment and household consumption expenditure, which are well-established indicators of SES and reflect different dimensions of socioeconomic status. We have also acknowledged the potential confounding effect of SES in the limitations section of the manuscript. Finally, in accordance with your suggestion, we have explicitly added a discussion in the discussion section regarding the impact of social and economic status on PAL and access to healthcare. For discussion additions, See line�320-331. For limitations additions, See line�360.

References on Education and Consumption Level as Indicators of Socioeconomic Status:

51. Broer M, Bai Y, Fonseca F. A Review of the Literature on Socioeconomic Status and Educational Achievement. In: Socioeconomic Inequality and Educational Outcomes. Springer; 2019:7-17. doi:10.1007/978-3-030-11991-1_2.

52. Poirier MJP. Systematic comparison of household income, consumption, and assets to measure health inequalities in low- and middle-income countries. Sci Rep. 2024;14(1):3851. E pub 20240215. Doi : 10.1038/s41598-024-54170-1. PubMed PMID: 38360925; PubMed Central PMCID: PMCPMC10869835.

(3)- Self-Reported CVD Diagnosis: Accuracy and Bias

The diagnosis of CVD is based on self-reported physician-diagnosed heart disease or stroke. This method is prone to recall bias and may lead to misclassification errors. Participants with undiagnosed CVD or those with subjective symptoms but no formal diagnosis may be misclassified. The sensitivity and specificity of self-reported CVD data should be addressed in the discussion, and comparisons with objectively measured CVD prevalence in similar populations should be provided if available.

Response:

In accordance with your suggestion, this study explicitly acknowledges the limitations associated with self-reported diagnoses of CVD. See line�361-364. However, Xie (2019). demonstrated that 77.5% of self-reported coronary heart disease cases were consistent with medical records in the English Longitudinal Study of Ageing (ELSA). Similarly Glymour (2009). found that misreporting of stroke was non-systematic, and self-reported stroke data could be reliably used to study the incidence and risk factors of stroke in the Health and Retirement Study (HRS). These findings suggest that the potential misclassification bias is likely minimal. Furthermore, self-reported CVD diagnoses have been widely used in various scientific studies.

Related references:

Xie W, Zheng F, Yan L., Zhong B. Cognitive decline before and after incident coronary events. J Am Coll Cardiol. 2019;24:3041–3050. Doi: 10.1016/j.jacc.2019.04.019.

Glymour M M, Avendano M. Can Self-Reported Strokes Be Used to Study Stroke Incidence and Risk Factors? [J/OL]. Stroke, 2009, 40(3): 873-879. DOI:10.1161/STROKEAHA.108.529479.

He D, Wang Z, Li J, Yu K, He Y, He X, et al. Changes in frailty and incident cardiovascular disease in three prospective cohorts. Eur Heart J. 2024;45(12):1058-68. Doi : 10.1093/eurheartj/ehad885. PubMed PMID: 38241094.

Yang Y, Cao L, Xia Y, Li J. The effect of living environmental factors on cardiovascular diseases in Chinese adults: results from a cross-sectional and longitudinal study. Eur J Prev Cardiol. 2023;30(11):1063-73. Doi : 10.1093/eurjpc/zwac304. PubMed PMID: 3653.

The prevalence of CVD has been compared in the Discussion section. Our study found an 11.86% prevalence of CVD among middle-aged and elderly populations, which is similar to the findings of Yuxiang W (2023). For discussion additions, See line�257-266.

Yuxiang W, Rulin M, Heng G, Xianghui Z, Jia H, Xinping W, et al. Prevalence of cardiovascular disease and its influencing factors among the Uygur population in rural areas of Xinjiang. Chinese Journal Of Disease Control & Prevention. 2023;27(4):385-91. Doi : 10.16462/j.cnki.zhjbkz.2023.04.003.

(4) PAL Categorization and MET-Minute Thresholds

The classification of low, moderate, and high PAL is based on arbitrary MET-minute cutoffs (<600, 600-3000, >3000 MET-min/week). However, it is unclear whether these categories align with established guidelines such as WHO, ACSM, or AHA recommendations. Justification should be provided for the chosen MET cutoffs. Alternatively, the analysis could be repeated using quartiles or deciles of PAL to explore whether different thresholds yield different results. The interpretation of 3000 MET-min/week as a threshold for reduced CVD prevalence should be critically examined, as it does not align directly with WHO recommendations for physical activity.

Response:

MET-minute/week cutoffs

The MET-minutes/week cutoffs (<600, 600–3000, and >3000 MET-min/week) were selected based on the classification used in the International Physical Activity Questionnaire (IPAQ). This method of physical activity categorization has been widely adopted in various studies. Below are references to studies that have utilized this classification.

In studies conducted by Lear (2017), Bauman (2009), and Dąbrowska (2019) et al., physical activity levels were assessed using the International Physical Activity Questionnaire (IPAQ) and categorized into low, moderate, and high, consistent with the classification method used in this study. Lear (2017) found that moderate and high physical activity levels were associated with a graded reduction in cardiovascular disease (CVD) mortality compared to low levels, while Dąbrowska et al. reported that women with high and moderate activity levels experienced milder menopausal symptoms compared to inactive counterparts.

Related references:

Lear SA, Hu W, Rangarajan S, et al. The effect of physical activity on mortality and cardiovascular disease in 130 000 people from 17 high-income, middle-income, and low-income countries: the PURE study published correction appears in Lancet. 2017 Dec 16;390(10113):2626. doi: 10.1016/S0140-6736(17)32596-5.

Bauman A, Bull F, Chey T, et al. The International Prevalence Study on Physical Activity: results from 20 countries. Int J Behav Nutr Phys Act. 2009;6:21. Published 2009 Mar 31. doi:10.1186/1479-5868-6-21.

Dąbrowska-Galas M, Dąbrowska J, Ptaszkowski K, Plinta R. High Physical Activity Level May Reduce Menopausal Symptoms. Medicina (Kaunas). 2019;55(8):466. Published 2019 Aug 11. doi:10.3390/medicina55080466.

PAL quartile

We conducted supplementary multiple logistic regression analyses using quartiles of PAL, and the results showed trends consistent with those observed in the three-category classification of physical activity. See line: 217.

Baseline characteristics of physical activity quartile and comparisons of differences between groups can be found in Supplementary File S1.

To 3000 MET-min/week as a threshold

In Model 2, after adjusting for hypertension, dyslipidemia, diabetes, drinking status, smoking status, both the moderate PAL (600-3000 MET minutes/week) and high PAL groups exhibited lower risk levels, whereas in the previous model, only the high PAL group (>3000 MET minutes/week) showed a lower risk. These findings align with the results reported by Lear et al. (2017).

Lear et al(2017). physical activity levels were assessed using the International Physical Activity Questionnaire (IPAQ) across 14 countries with varying economic statuses. Total physical activity was categorized into low (<600 MET-minutes/week), moderate (600–3000 MET-minutes/week), and high (>3000 MET-minutes/week). The study found that moderate and high physical activity levels were associated with a graded reduction in cardiovascular disease (CVD) mortality compared to low physical activity levels (<600 MET-minutes/week).

Lear SA, Hu W, Rangarajan S, et al. The effect of physical activity on mortality and cardiovascular disease in 130 000 people from 17 high-income, middle-income, and low-income countries: the PURE study published correction appears in Lancet. 2017 Dec 16;390(10113):2626. Doi: 10.1016/S0140-6736(17)32596-5.

The findings of our study partially align with the WHO and ACSM recommended physical activity levels of 600-1200 MET-minutes/week. The discrepancy may arise because the physical activity levels recommended by WHO and ACSM refer to the amount of activity required in addition to daily lifestyle activities, whereas the physical activity levels in this study include non-exercise daily physical activities. Furthermore, most current studies suggest that exceeding the physical activity levels recommended by the WHO and ACSM guidelines is more beneficial than merely meeting these recommendations.

Williams PT. Dose-response relationship of physical activity to premature and total all-cause and cardiovascular disease mortality in walkers. PLoS One. 2013;8(11):e78777. Published 2013 Nov 29. doi:10.1371/journal.pone.0078777

Thompson D, Batterham AM, Peacock OJ, Western MJ, Booso R. Feedback from physical activity monitors is not compatible with current recommendations: A recalibration study. Prev Med. 2016; 91: 389-394.doi: 10.1016/j. ypmed. 2016.06.017.

(5)- Restricted Cubic Spline Interpretation

The restricted cubic spline analysis suggests a linear inverse association between PAL and CVD (p>0.05 for non-linearity). However, Figure 2 should be more clearly annotated to indicate where the confidence intervals widen, which could suggest statistical uncertainty at extreme PAL levels. The authors should discuss whether this linear trend holds across all PAL levels or whether the association plateaus at higher MET levels, as some studies suggest diminishing returns beyond a certain point.

Response:

Figure 2 caption

In the Figure 2, the OR values are represented by the solid red line, with the 95% confidence intervals (95% CI) indicated by the shaded gray area. the overall trend of the curve demonstrates a negative correlation, with OR values decreasing as amount of physical activity increase. This indicates that higher levels of physical activity are associated with a lower risk of CVD prevalence. At weekly physical activity levels exceeding10,000 MET-minutes/week, the curve begins to plateau, suggesting that the additional health benefits of further increases in physical activity may diminish. Furthermore, the confidence intervals were narrow at moderate and high physical activity levels, indicating greater statistical stability. However, at extreme MET-minutes/week values (<600 or >12,000), the confidence intervals wide, reflecting increased uncertainty due to fewer data points within these ranges. For figure note additions, See line�227-235、238-243.

linear trend holds across all PAL levels

The restricted cubic spline plot revealed a significant inverse linear trend between weekly physical activity levels (ranging from 0 to 15,000 MET-minutes/week) and CVD risk. Specifically, higher levels of physical activity are consistently associated with a lower risk of CVD prevalence. The odds ratio demonstrated a linear decline as physical activity levels increased. However, the curve began to plateau when physical activity exceeded 10,000 MET-minutes/week. This phenomenon suggests the potential existence of a threshold effect in the relationship between physical activity and CVD risk. Specifically, further increases in physical activity beyond a certain level may not significantly contribute to additional reductions in CVD risk. This finding aligns with existing research, which indicates that the health benefits of physical acti

---

## [Decision Letter · Decision Letter 1]

PONE-D-24-53272R1Association between physical activity level and cardiovascular disease: an empirical analysis based on CHARLS data in 2018PLOS ONE

Dear Dr. Li,

Thank you for submitting your manuscript to PLOS ONE. After careful consideration, we feel that it has merit but does not fully meet PLOS ONE’s publication criteria as it currently stands. Therefore, we invite you to submit a revised version of the manuscript that addresses the points raised during the review process.

We look forward to receiving your revised manuscript.

Kind regards,

Amirmohammad Khalaji

Academic Editor

PLOS ONE

Reviewers' comments:

Reviewer's Responses to Questions

**Comments to the Author**

1. If the authors have adequately addressed your comments raised in a previous round of review and you feel that this manuscript is now acceptable for publication, you may indicate that here to bypass the “Comments to the Author” section, enter your conflict of interest statement in the “Confidential to Editor” section, and submit your "Accept" recommendation.

Reviewer #1: All comments have been addressed

Reviewer #2: (No Response)

2. Is the manuscript technically sound, and do the data support the conclusions?

Reviewer #1: Yes

Reviewer #2: Partly

3. Has the statistical analysis been performed appropriately and rigorously? 

Reviewer #1: Yes

Reviewer #2: No

4. Have the authors made all data underlying the findings in their manuscript fully available?

Reviewer #1: Yes

Reviewer #2: No

5. Is the manuscript presented in an intelligible fashion and written in standard English?

Reviewer #1: Yes

Reviewer #2: No

6. Review Comments to the Author

Reviewer #1: No comments as manuscript is throughly revised and all comments addressed as appropriate by authors. I applaud for their work

Reviewer #2: Weaknesses (Major issues):

Handling of residual confounding (especially socioeconomic status) is insufficiently robust.

The definition of PAL categories (low, moderate, high) remains somewhat arbitrary despite partial justification.

Some new additions (e.g., quartile analyses) are insufficiently integrated into the discussion.

The narrative on plateau effects in the dose-response relationship needs more depth.

Figure 2 still lacks enough annotation to guide readers in interpreting wide confidence intervals.

Detailed Comments

Introduction

Good improvement on causal language. However, some remaining sentences still sound too causal (e.g., “maintaining moderate-to-high physical activity reduces the burden of CVD”). Please revise consistently to “associated with”.

Methods

Exposure Measurement:

The reliance solely on self-reported PAL should be more critically discussed upfront. Objective measures (e.g., accelerometers) are unavailable, but the authors could mention this as a limitation in the Methods itself (not only in Discussion).

Missing Data Handling:

The handling of missing data was by case-wise deletion. Given the substantial exclusions (over 10,000 participants), please add a sensitivity analysis or at least discuss the potential for selection bias more explicitly in the Discussion section.

Covariate Selection:

While education and household consumption were used as proxies for socioeconomic status (SES), the absence of an explicit SES composite variable is a concern. Discuss this more clearly. Alternatively, re-analyze with household consumption quintiles or categories if possible.

Results

Figure 2 (Restricted Cubic Spline):

While the figure is technically correct, the wide confidence intervals at high PAL (>12,000 MET-min/week) and low PAL (<600 MET-min/week) need explicit labels or annotations.

Suggest adding dotted vertical lines at 600, 3000, and 10,000 MET-minutes/week to show thresholds.

Briefly describe in the Results section when the OR curve begins to plateau (currently mixed into the Discussion only).

Discussion

Residual Confounding:

The potential for residual confounding (especially unmeasured variables like diet, medication adherence, genetic factors) should be acknowledged.

Current references suggesting minimal misclassification for self-reported CVD are appreciated but still optimistic; please explicitly note that this might not fully apply to the Chinese CHARLS cohort.

Plateau Effect:

You mention a plateau at >10,000 MET-min/week, but the biological plausibility and comparison to other cohort studies (e.g., PURE, UK Biobank) should be discussed.

Also, briefly speculate on whether very high levels of activity might confer risks (overtraining, cardiovascular stress).

Generalizability:

Good discussion on urban/rural differences and cultural factors. It would be helpful to strengthen the point that findings might not apply to populations where leisure-time physical activity (rather than occupational activity) predominates.

Minor Points

Proofreading is still needed:

Fix minor English grammar issues (e.g., "the confidence intervals wide" → "the confidence intervals widened").

Consistent tenses (prefer "was/were" rather than "is/are" when describing past data).

7. PLOS authors have the option to publish the peer review history of their article (what does this mean? ). If published, this will include your full peer review and any attached files.

**Do you want your identity to be public for this peer review?** For information about this choice, including consent withdrawal, please see our Privacy Policy .

Reviewer #1: No

Reviewer #2: No

---

## [Author Response · Author response to Decision Letter 2]

21 May 2025

Response to reviewers

We are so grateful for the thoughtful reviews provided by the external referees. We have tried to carefully address each recommendation. We believe that our manuscript is now stronger as a result of the modifications outlined below. We also thank the editoral staff and peer-reviewers for their contributions.

Revisions in the manuscript are highlighted in yellow. In addition to the specific responses to reviewers below, we have made revisions to grammar and tense based on suggestions from the reviewers and editors.

Reviewer #2

Weaknesses (Major issues)

(1) -Handling of residual confounding (especially socioeconomic status) is insufficiently robust.

Response We sincerely appreciate the reviewer's insightful comments regarding potential residual confounding related to socioeconomic status (SES). In line with current methodological standards in the literature, we have reconceptualized the assessment of socioeconomic status (SES) through

Socioeconomic status was comprehensively assessed through two key dimensions: educational attainment and per capita household expenditure. First, participants' highest educational level was quantified using standardized values (illiterate/semiliterate = 0, sishu/elementary school = 6, middle school = 9, high school/vocational school = 12, college = 16, postgraduate = 19). Second, per capita expenditure was calculated by dividing total household expenditure by the number of household members. To account for scale differences, both education level and per capita expenditure were standardized using Z-scores, and these standardized values were summed to create a composite SES score, with higher values indicating higher SES status. Participants were then stratified into three groups using tertile categorization: low SES (bottom tertile), medium SES (middle tertile), and high SES (top tertile), ensuring balanced group sizes. See line�159-169.

The robustness of our primary findings persisted following SES covariate adjustment (PAL�[Model 2: Moderate PALs: OR=0.79,95% CI: 0.64-0.97, P<0.05, High PALs: OR=0.72, 95% CI: 0.60~0.87, P<0.05]�PAL quartile�[Model 2�Q2:1733~4158,OR=0.90,95% CI�0.75-1.07�P=0.225�Q3:4158~9198,OR=0.78,95% CI�0.65-0.94�P<0.05�Q4:>9198,OR=0.64,95% CI�0.52-0.79�P<0.001]) Moreover, in accordance with your recommendations, we have explicitly acknowledged in the Limitations section the potential for residual confounding arising from unmeasured factors including, but not limited to: diet, medication adherence, genetic factors. See line�216-218、395.

Socioeconomic Status References:

Li W, Zhang X, Gao H, Tang Q. Heterogeneous effects of socio-economic status on social engagement level among Chinese older adults: evidence from CHARLS 2020. Front Public Health. 2024;12:1479359. Published 2024 Nov 29. doi:10.3389/fpubh.2024.1479359

(2) -The definition of PAL categories (low, moderate, high) remains somewhat arbitrary despite partial justification. Some new additions (e.g., quartile analyses) are insufficiently integrated into the discussion.

Response We sincerely appreciate your valuable feedback on the research methodology and integration of results. We fully agree with your perspective that the classification of physical activity levels (PAL) requires more thorough justification and have made the following improvements to address the insufficient incorporation of the new quartile analysis in the discussion section.

Currently, there are multiple classification methods for physical activity levels. In this study, we adopted a grouping approach based on existing literature, which may still appear somewhat arbitrary. Notably, following your suggestion, we conducted supplementary quartile analyses, which revealed a more nuanced dose-response relationship between physical activity levels (PAL) and CVD risk. As physical activity increased from Q1 (<1,733 MET-minutes/week) to Q4 (>9,198 MET-minutes/week), the odds ratios (ORs) for CVD risk demonstrated a clear graded decline (OR decreasing from 0.90 to 0.64, P for trend <0.05). This monotonic decreasing trend further supports the protective effect of physical activity. Specifically, compared to the least active group (Q1), the Q2 group (1,733–4,158 MET-minutes/week) showed a 10% reduction in risk (OR = 0.90, 95% CI: 0.75–1.07), the Q3 group (4,158–9,198 MET-minutes/week) exhibited a significant 22% decrease (OR = 0.78, 95% CI: 0.65–0.94), and the Q4 group demonstrated the greatest reduction at 36% (OR = 0.64, 95% CI: 0.52–0.79). Although the three-category and quartile analyses used different cutoff points, both approaches consistently demonstrated a similar decreasing trend." This concordance across models strengthens the credibility of our findings. These results provide valuable insights for developing more precise exercise recommendations, suggesting that exceeding minimum activity guidelines may yield additional health benefits. However, optimal activity thresholds may vary across populations. See line�282-292.

Additionally, newly incorporated covariates—including hypertension, diabetes, dyslipidemia, smoking, alcohol consumption, and socioeconomic status—were integrated into the discussion, as these factors are also strongly associated with CVD risk. See line�274-279.

(3) -Plateau Effect The narrative on plateau effects in the dose-response relationship needs more depth. You mention a plateau at >10,000 MET-min/week, but the biological plausibility and comparison to other cohort studies (e.g., PURE, UK Biobank) should be discussed. Also, briefly speculate on whether very high levels of activity might confer risks (overtraining, cardiovascular stress).

Response From a biological mechanism perspective, the health benefits of physical activity typically follow a nonlinear dose-response curve. Regular, moderate exercise can significantly reduce the risk of various chronic diseases by improving insulin sensitivity, reducing systemic inflammation, and enhancing cardiopulmonary function. However, when physical activity exceeds a certain threshold, the body's metabolic and organ systems reach relative saturation, potentially leading to diminishing health returns. This manifests as a plateau effect where the magnitude of risk reduction gradually stabilizes. See line�310-316.

A large cohort study from PURE (Prospective Urban Rural Epidemiology) demonstrated graded reductions in mortality (HR 0.80, 95% CI 0.74-0.87 and 0.65, 0.60-0.71; p-trend<0.0001) and major CVD (0.86, 0.78-0.93; p-trend<0.001) when comparing moderate (600-3000 MET-minutes/week) and high (>3000 MET-minutes/week) activity levels against low activity (<600 MET-minutes). Another UK Biobank study identified significant linear dose-response relationships between moderate/vigorous/total physical activity levels and CVD risk. Additionally, the study by Bakker EA et al, identified a dose-response relationship between PAL, major adverse cardiovascular events (MACE), and all-cause mortality across different population groups: healthy individuals, individuals with cardiovascular risk factors(CVRF),and individuals with CVD. PAL, measured in MET-minutes per week, was divided into quartiles: Q1 (1–1912 MET-minutes/week), Q2 (1913–3690 MET-minutes/week), Q3 (3691–7257 MET-minutes/week), and Q4 (>7258 MET-minutes/week). The study found that higher PAL was associated with a lower risk of adverse outcomes in both healthy individuals and patients with CVRF. For CVD patients, only the highest PA quartile was associated with a lower risk of adverse prognosis. See line�319-333.

However, extremely high activity levels may not be absolutely safe. Research indicates that sustained excessive exercise may trigger overtraining syndrome, potentially inducing learning and memory impairments through elevated inflammatory cytokines and oxidative stress markers. Additionally, studies report that chronic endurance athletes may develop atrial dilation and increased arrhythmia risk. In summary, physical activity exceeding 10,000 MET-minutes/week may reach a "physiological saturation point" - while not immediately hazardous, the health benefits plateau. Beyond this threshold, certain individuals may experience diminishing returns or potential adverse effects with further training load escalation. See line�339-346.

(4) -Figure 2 still lacks enough annotation to guide readers in interpreting wide confidence intervals.

Response As per your suggestion, a detailed figure legend has been added below Figure 2 to help readers understand the wide confidence intervals.

Figure caption: Association between amount of physical activity and cardiovascular disease risk. This figure presents the results of RCS analysis based on Model 2, illustrating the dose-response relationship between physical activity (measured in MET-minutes/week) and the OR for cardiovascular disease. The OR values are represented by the solid red line, with the 95% confidence intervals (95% CI) indicated by the shaded gray area. The graph is truncated at 15,000 MET-minutes/week, corresponding to the 93rd percentile of the physical activity distribution. The red vertical dashed lines indicated key thresholds at 600, 3,000, 10,000, and 12,000 MET-minutes per week. The confidence intervals were wider at low (<600 MET-minutes/week) and high (>12,000 MET-minutes/week) levels of physical activity, which may be due to smaller sample sizes or greater data variability at these extremes. In contrast, within the 3,000–10,000 MET-minutes/week range, the sample size was larger and more stable, resulting in narrower confidence intervals. Overall, the trend showed that the risk ratio decreased progressively with increasing physical activity. See line�245-251.

Detailed Comments

Introduction

Good improvement on causal language. However, some remaining sentences still sound too causal (e.g., “maintaining moderate-to-high physical activity reduces the burden of CVD”). Please revise consistently to “associated with”.

Response Thank you for your careful review and for pointing this out. We are grateful for your acknowledgment of the improvements made and fully agree on the importance of avoiding causal language in observational studies. Accordingly, we have thoroughly examined the manuscript's introduction section and revised all remaining instances that might contain causal implications. For example, the original statement: 'Research shows that, compared to participants reporting no leisure-time physical activity, those who engaged in guideline-recommended minimum levels of physical activity—500 MET-minutes/week according to the 2008 US guidelines—experienced modest lower CVD risk' has been modified to: 'Research shows that, compared to participants reporting no leisure-time physical activity, those meeting the minimum physical activity level recommended by the 2008 U.S. guidelines (500 MET-minutes/week) were associated with a lower CVD risk.' All similar expressions throughout the manuscript have been corrected to reflect associations rather than causal relationships." See line�76-82.

Methods

Exposure Measurement:

The reliance solely on self-reported PAL should be more critically discussed upfront. Objective measures (e.g., accelerometer) are unavailable, but the authors could mention this as a limitation in the Methods itself (not only in Discussion).

Response According to your suggestion, this limitation has been added in the methodology section: PAL is assessed through self-report questionnaires. However, Objective tools such as accelerometer were not employed, which may introduce potential bias due to inaccurate recall or reporting. See line�111-112.

Missing Data Handling:

The handling of missing data was by case-wise deletion. Given the substantial exclusions (over 10,000 participants), please add a sensitivity analysis or at least discuss the potential for selection bias more explicitly in the Discussion section.

Response To verify the robustness of our primary findings, we conducted sensitivity analyses by reclassifying physical activity levels (PAL) from the original three-category grouping (low/medium/high) into quartiles (Q1-Q4), as suggested by your advice. The results demonstrated consistent trends between the quartile-based and three-category analyses (Table 3), supporting the reliability of our main findings. We have incorporated the discussion of these quartile-based analyses in the manuscript. See line�282-292. Additionally, we have included a discussion of potential selection bias in the limitations section to more explicitly address the possibility of such bias. See line�399-402.

Covariate Selection:

While education and household consumption were used as proxies for socioeconomic status (SES), the absence of an explicit SES composite variable is a concern. Discuss this more clearly. Alternatively, re-analyze with household consumption quintiles or categories if possible.

Response Following your recommendation and consistent with established methodological standards in the literature, we redefined socioeconomic status (SES) and included it as a covariate in the logistic regression analyses. The robustness of our primary findings remained after SES adjustment, with full results detailed in Response to Comment 1. See line�159-169.

Results

Figure 2 (Restricted Cubic Spline):

While the figure is technically correct, the wide confidence intervals at high PAL (>12,000 MET-min/week) and low PAL (<600 MET-min/week) need explicit labels or annotations. Suggest adding dotted vertical lines at 600, 3000, and 10,000 MET-minutes/week to show thresholds. Briefly describe in the Results section when the OR curve begins to plateau (currently mixed into the Discussion only).

Response As requested, we have added dashed lines at 600, 3,000, 10,000, and 12,000 MET-minutes/week in Figure 2 to indicate threshold values, along with clear labels and annotations in the figure legend. Additionally, we have included a description in the Results section specifying when the OR curve begins to plateau. See line�245-251、231-233.

Discussion

Residual Confounding:

The potential for residual confounding (especially unmeasured variables like diet, medication adherence, genetic factors) should be acknowledged.

Response Per your suggestion, we have added diet, medication adherence, and genetic factors as potential residual confounders in the Limitations section. See line�395.

Current references suggesting minimal misclassification for self-reported CVD are appreciated but still optimistic; please explicitly note that this might not fully apply to the Chinese CHARLS cohort.

Response As per your suggestion, we have added the following content to the manuscript: Although prior studies indicate a relatively low misclassification rate for self-reported CVD, these findings should be interpreted with caution given the substantial proportion of rural participants in the CHARLS cohort in China, along with their limited CVD awareness and potential recall bias, which may increase misclassification risk. See line�413-416.

Plateau Effect:

You mention a plateau at >10,000 MET-min/week, but the biological plausibility and comparison to other cohort studies (e.g., PURE, UK Biobank) should be discussed.

Also, briefly speculate on whether very high levels of activity might confer risks (overtraining, cardiovascular stress).

Response Per your suggestion, we have added the following content to the Discussion section: (1) the biological mechanisms underlying our observational findings, (2) comparisons with results from other cohort studies, and (3) preliminary speculation about potential adverse effects associated with extremely high physical activity levels. Please see Response 3 for details. See line�310-316、319-333、339-346.

Generalizability:

Good discussion on urban/rural differences and cultural factors. It would be helpful to strengthen the point that findings might not apply to populations where leisure-time physical activity (rather than occupational activity) predominates.

Response We sincerely appreciate the reviewer's positive feedback regarding our discussion of urban-rural disparities and cultural factors. We fully concur with the insightful observation that our findings may not be entirely generalizable to populations where leisure-time physical activity p

---

## [Decision Letter · Decision Letter 2]

Association between physical activity level and cardiovascular disease: an empirical analysis based on CHARLS data in 2018

PONE-D-24-53272R2

Dear Dr. Li,

We’re pleased to inform you that your manuscript has been judged scientifically suitable for publication and will be formally accepted for publication once it meets all outstanding technical requirements.

Kind regards,

Amirmohammad Khalaji

Academic Editor

PLOS ONE

Additional Editor Comments (optional):

Reviewers' comments:

Reviewer's Responses to Questions

**Comments to the Author**

1. If the authors have adequately addressed your comments raised in a previous round of review and you feel that this manuscript is now acceptable for publication, you may indicate that here to bypass the “Comments to the Author” section, enter your conflict of interest statement in the “Confidential to Editor” section, and submit your "Accept" recommendation.

Reviewer #2: All comments have been addressed

2. Is the manuscript technically sound, and do the data support the conclusions?

Reviewer #2: Yes

3. Has the statistical analysis been performed appropriately and rigorously? 

Reviewer #2: Yes

4. Have the authors made all data underlying the findings in their manuscript fully available?

Reviewer #2: Yes

5. Is the manuscript presented in an intelligible fashion and written in standard English?

Reviewer #2: Yes

6. Review Comments to the Author

Reviewer #2: This manuscript represents a valuable and well-conducted epidemiological study examining the association between physical activity and cardiovascular disease in a large, nationally representative Chinese cohort. The authors have addressed prior reviewer comments in depth, improving the manuscript substantially. The analyses are robust, the discussion is thoughtful, and the public health relevance is clear. While limitations remain—especially the cross-sectional design and self-reported data—the strengths outweigh these issues.

7. PLOS authors have the option to publish the peer review history of their article (what does this mean? ). If published, this will include your full peer review and any attached files.

**Do you want your identity to be public for this peer review?** For information about this choice, including consent withdrawal, please see our Privacy Policy .

Reviewer #2: No

---

## [Editor Report · Acceptance letter]

PONE-D-24-53272R2

PLOS ONE

Dear Dr. Li,

I'm pleased to inform you that your manuscript has been deemed suitable for publication in PLOS ONE. Congratulations! Your manuscript is now being handed over to our production team.

Kind regards,

on behalf of

Dr. Amirmohammad Khalaji

Academic Editor

PLOS ONE